# Methods for Radiolabelling Nanoparticles: SPECT Use (Part 1)

**DOI:** 10.3390/biom12101522

**Published:** 2022-10-20

**Authors:** Michela Varani, Valeria Bentivoglio, Chiara Lauri, Danilo Ranieri, Alberto Signore

**Affiliations:** 1Nuclear Medicine Unit, Department of Medical-Surgical Sciences and of Translational Medicine, Faculty of Medicine and Psychology, “Sapienza” University of Rome, 00189 Roma, Italy; 2Department of Clinical and Molecular Medicine, Faculty of Medicine and Psychology, “Sapienza” University of Rome, 00189 Roma, Italy

**Keywords:** nanoparticles, nanotechnology, nuclear medicine, radiolabelling, SPECT/CT

## Abstract

The use of nanoparticles (NPs) is rapidly increasing in nuclear medicine (NM) for diagnostic and therapeutic purposes. Their wide use is due to their chemical–physical characteristics and possibility to deliver several molecules. NPs can be synthetised by organic and/or inorganic materials and they can have different size, shape, chemical composition, and charge. These factors influence their biodistribution, clearance, and targeting ability in vivo. NPs can be designed to encapsulate inside the core or bind to the surface several molecules, including radionuclides, for different clinical applications. Either diagnostic or therapeutic radioactive NPs can be synthetised, making a so-called theragnostic tool. To date, there are several methods for radiolabelling NPs that vary depending on both the physical and chemical properties of the NPs and on the isotope used. In this review, we analysed and compared different methods for radiolabelling NPs for single-photon emission computed tomography (SPECT) use.

## 1. Introduction

In recent years, the synthesis of nanoparticles (NPs) as innovative nanoradiopharmaceuticals has demonstrated great potential for a future translation in clinical applications. NPs exhibit unique chemical and physical characteristics that allow to consider them a solution for several unsolved issues in clinical settings. The availability of different materials to form the structure of the NPs, with controlled shape and size, high stiffness, discrete charge, or electromagnetic properties, combined with their high surface-to-volume ratio properties and the tuneable reactivity of their surface, allow us to customize the design of the NPs for specific biomedical applications [1]. Several procedures to radiolabel NPs have been published so far, using different methods depending on the radioisotope and the characteristics of the NPs. The radioisotope can be selected based on its nuclear decay, mode of production, and radiolabelling method. During the radiolabelling procedure, the match between the biological half-life of NPs and the half-life of the isotope is one of the most important considerations to allow the visualisation of the NPs until the target is reached and to avoid unnecessary radiation exposure. Besides the use of radionuclides for imaging and/or therapy, certain combinations of radioisotopes can be used as theragnostic pairs for both imaging and therapy. These combinations are composed by two radioisotopes of the same chemical element, one with the appropriate radiophysical properties to generate a signal for PET or SPECT detection, and the other isotope with suitable therapeutic properties. This is an interesting approach, since both isotopes can be of the same element and hence do not require different labelling methods or chelators. Similarly, isotopes can be different for diagnostic or therapeutic purposes but belong to the same chemical group, thus requiring a different labelling method but the same chelator [2]. The selection of a suitable NP for radiolabelling is based on its clinical application. It is important to consider the nanomaterial of which it is composed, the presence of reactive groups on the surface, the hydrodynamic diameter, and the charge. NPs may be labelled by means of a direct or indirect method, as well as by encapsulating the radioisotopes (Figure 1). The direct labelling to the surface of NPs does not include the chelator or spacers that are, conversely, needed in indirect method [2]. Bifunctional chelators or spacers are mainly used to label metal radionuclides. In this case, the choice of a specific chelator depends on the oxidation state of the radiometal and on its nature [3]. The encapsulation of the isotope could be performed during the synthesis of NPs or by a postsynthesis method. Independently, by the method, the radiolabelling procedure could affect the physical or chemical properties of the NPs. Therefore, it is necessary to investigate the size, zeta potential, and surface properties of NPs after radiolabelling.

Other important factors to consider after the radiolabelling of NPs are the labelling efficiency (LE), the recovery yield (RCY), and the specific activity (SA).

LE is the percentage of radioactivity labelled to NPs divided the total amount added to the solution. High LE (over 95%) does not require the further purification of the NP suspensions to eliminate the free isotope present in solution. The RCY represents the ratio between the final amount of radioactivity labelled to the NPs and the starting activity used for the radiolabelling. Thus, a high RCY indicates a reduced loss of radioactivity in the radiolabelling process.

SA indicates the amount of radioactivity per gram or mole of NPs. High SA involves a small quantity of nanomaterials. This is important both for diagnostic and therapeutic studies, since the use of a small quantity of NPs reduces the side or other biological effects in vivo [4].

In general, the radiolabelling process should not alter the physical and chemical properties of NPs, and all these parameters should be analysed during the formulation of new nanoradiopharmaceuticals.

This review provides an updated overview on the main single-photon emitting computed tomography (SPECT) isotopes and methods used for radiolabelling NPs.

## 2. Radiolabelling of NPs for SPECT Imaging

Radiolabelling has been performed with several isotopes for SPECT imaging. As previously mentioned, the techniques can be divided into three main methods: direct labelling, indirect labelling, and methods for isotope encapsulation.

### 2.1. Radiolabelling with Techentium-99m

The most common SPECT isotope used in diagnostic nuclear medicine (NM) is ^99m^Tc due to its low cost and high availability from ^99^Mo/^99m^Tc generators, its short half-life (6 h), and excellent energy (140.5 keV). Furthermore, the highly stable coordination chemistry makes it an ideal radioisotope for high-purity radiopharmaceutical production. The favourable chemical properties and easy labelling process of ^99m^Tc have led to an increase in its application for NP radiolabelling in recent years [5,6].

#### 2.1.1. Direct Radiolabelling

The direct radiolabelling of NPs with ^99m^Tc involves the use of a reducing agent, usually an acidic stannous chloride (SnCl_2_) solution, to reduce the heptavalent oxidation state to lower oxidation states, forming a metastable specie and facilitating the labelling on the NP surface [7,8,9,10].

The main limitation of this approach is the creation of colloidal ^99m^Tc stannic oxide (TcO_2_) generated during the reduction procedure. These colloidal particles are difficult to purify from NPs with a size of 50–200 nm due to their similar size. Moreover, if injected, the radiocolloids would concentrate in the reticuloendothelial cells (RES), thus interfering with the normal biodistribution of the NPs [11,12]. Therefore, the percentage of radiocolloids should be always assessed.

The amount of SnCl_2_ is relevant for the LE, since, if its concentration is lower than the optimum value, an excess of unlabelled ^99m^TcO_4_^-^ may be observed, whereas a higher concentration exposes it to the risk of radiocolloid creation. The LE and RCY are mainly calculated with ascending instant thin-layer chromatography (ITLC) using silica gel (SG)-coated strips. The presence of free ^99m^TcO_4_^-^ in the solution is calculated with sodium chloride (NaCl 0.9%) or acetone or ethanol–water (7:3) as mobile phases, allowing its migration to the front of the SG strips with a retention factor (RF) between 0.8 and 1.0 and leaving the colloids and the ^99m^Tc-NPs at the origin. When a solution of pyridine–acetic acid–water (3:5:1.5) is used as the mobile phase, the radiocolloids remain at the origin of the SG strips, and the free ^99m^TcO_4_^-^ and radiolabelled NPs migrate to the front, allowing to calculate the percentage of radiocolloids.

Other different reducing agents were studied to avoid the colloidal formation in the direct labelling of NPs with ^99m^Tc. Geskovski et al. compared SnCl_2_ with sodium borohydride (NaBH_4_) and sodium dithionite (Na_2_S_2_O_4_) for the labelling of PLGA NPs with ^99m^Tc. By using SnCl_2_, the presence of colloidal ^99m^Tc stannic oxide was observed, thus interfering with the LE. The NaBH_4_ showed lower complex stability, resulting in high thyroid and gastric uptake in vivo. Na_2_S_2_O_4_ showed the formation of a stable complex of ^99m^Tc-PLGA with no colloidal precipitates, long blood circulation, and low hepatic absorption in vivo [13].

Na_2_S_2_O_4_ as a reducing agent for ^99m^Tc in the direct radiolabelling of chitosan NPs was also used by Farrag et al., reporting that 10 mg of Na_2_S_2_O_4_ were able to convert 7.2 MBq of ^99m^TcO_4_^−^ into the reduced form, with a final LE of 93.4 ± 1.2% [14].

Ashraf et al. demonstrated that the direct radiolabelling of silver NPs with ^99m^Tc and NaBH_4_ as reducing agent resulted in a LE% of more than 80% [15]. NaBH_4_ was also compared to SnCl_2_ for the radiolabelling of chitosan NPs. Both reducing agents were able to form a complex between the ^99m^Tc and the NPs, with a LE of 90% with SnCl_2_ and of 80–85% with NaBH_4_. The stability of the labelling was also higher for SnCl_2_. However, in vivo results confirmed the presence of colloids, with a high uptake in the liver, spleen, and lungs, while, by using the NaBH_4_ as reducing agent, ^99m^Tc-NPs showed lower accumulation in the RES organs and a longer residence time in blood circulation [16].

Another easy and fast method to avoid the formation of colloids was attempted by Tassano et al. by using the organometallic ^99m^Tc carbonyl precursor [^99m^Tc(CO)_3_(H2O)_3_]^+^ for the radiolabelling of dendrimers. The amines groups of dendrimers strongly labelled the tricarbonyl core of the precursor, with a final high LE (<90%) and high stability up to 24 h [17].

Other parameters, such as pH, temperature, NP concentration, and incubation time, should be considered to achieve the best LE.

The pH of the solution during the radiolabelling process is important to obtain an appropriate reduction of ^99m^Tc without affecting the NP properties. The required range is in the neutral pH area (pH:6–8) and sodium boranocarbonate is usually used as buffer [18]. The influence of incubation time, pH, and the amount of SnCl_2_ was assessed by Mirkovic et al. for the radiolabelling of bisphosphonate-coated magnetic nanoparticles (Fe_3_O_4_ MNPs). They coated Fe_3_O_4_ with two hydrophilic bisphosphonate ligands, methylene diphosphonate (MDP) and 1–hydroxyethane-1,1- diphosphonate (HEDP). Both of these formulations were radiolabelled adding ^99m^Tc and SnCl_2_ simultaneously to the NP suspension, showing a high RCY. The effect of the pH and the incubation time was assessed by changing one parameter at a time; in particular, the pH was adjusted from 4 to 7, and the reaction mixture was gently stirred and incubated at room temperature from 15 to 60 min. The radiolabelling process was completed after 30 min but increasing the incubation time up to 60 min would result in a slightly lower LE. On the other hand, the variation of pH (from 4 to 7) did not significantly affect the final LE. They showed that the crucial factor for the LE is the amount of SnCl_2_. The quantity of the reduction agent was varied from 275 to 400 µg, and the impact on the LE differed for the two different NPs. Indeed, the high LE for Fe_3_O_4_–MDP MNPs was obtained with 275 μg of SnCl_2_, and for Fe_3_O_4_–HEDP MNPs with 400 μg of SnCl_2_ [19]. This study demonstrated that NPs with the same physical properties, but functionalised with different ligands, can have a different radiolabelling process with a different final LE. Therefore, it is very important to study the different parameters that can influence the radiolabelling process so to obtain the highest possible LE for that type of NP.

The direct radiolabelling of NPs with ^99m^Tc can be also performed with a previous reduction of isotope by SnCl_2_, and the following incubation with NPs suspension. The method could be useful for optimising the amount of the reducing agent and the pH of solution before the conjugation with the NPs [20,21,22,23].

Pinto et al. radiolabelled magnetic core mesoporous silica NPs with ^99m^Tc, previously reduced with SnCl_2_. The quality controls showed a LE of >98% and no significant dissociation of the radioisotope from the NPs up to 8 h [24].

Following the same method, Eroglu et al. radiolabelled solid lipid NPs (SLNs) by first reducing ^99m^Tc with SnCl_2,_ and then adding this complex to an NP solution at room temperature for 10 min. The LE resulted greater than 95% [25]. The radiolabelling of the NPs with a direct method can also be performed with a short incubation between SnCl_2_ and the NPs, followed by the addition of ^99m^Tc in the solution. This method was used to radiolabel several types of NPs: octreotide NPs, cerium oxide NPs (CeO_2_-NPs), mesoporous silica NPs, poly-γ-glutamic acid NPs conjugated with folic acid, carbon-based mesoporous NPs, zirconium NPs, serum albumin nanoparticles, polymeric NPs, and dendrimer glycoconjugate [26,27,28,29,30,31,32,33,34]. All the studies showed a high final LE, but the formation of radiocolloids often requires further purification of the solution. Magnetic NPs, such as SPIONs, can be easily purified through the magnetic retraction by using their physical properties, allowing for a high LE and RCY [35,36]. The addition of ^99m^Tc at the final step of the labelling procedure has the great advantage to reduce the operator’s exposure.

Suchánková et al. compared two different approaches to radiolabelled hydroxyapatite NPs.

The first was a direct labelling, where the radionuclide was directly conjugated with the surface of previously synthetised NPs. NPs were dispersed in fresh SnCl_2_ solution and then ^99m^Tc was added. The samples were mixed for one hour at room temperature and then washed three times with saline.

The second strategy was an intrinsic labelling: the radionuclide was incorporated directly into the NPs’ structure by adding Ca(NO_3_)_2_ in demineralised water. Then, a small volume of ^99m^Tc solution was added. (NH_4_)_2_HPO_4_ was added and the mixture was stirred for 1 h at room temperature. Finally, the samples were washed with saline. Both the methods showed a high RCY (≥94%), thus demonstrating that they are both feasible strategies for the labelling [37].

Ge et al. developed a new radiolabelling method called “ligand anchoring group-mediated radiolabelling” (LAGMERAL) based on the coordination between diphosphonate–polyethylene glycol (DP-PEG) ligands present on the surface of inorganic NPs and the metal radioisotope. This method showed an excellent colloidal stability and an efficient labelling through the diphosphonate group, making the NPs water-soluble and biocompatible. In this study, three types of NPs were radiolabelled with ^99m^Tc: Fe_3_O_4_, NaGdF_4_, and Cu_2_−xS NPs. The synthesis of Fe_3_O_4_ NPs was performed by the thermal decomposition of ferric acetylacetonate, followed by the exchange of the hydrophobic oleate ligand with the PEG ligand, with a diphosphonate group at one end and a methoxy group at the other end.

The PEGylated Fe_3_O_4_ NPs were dispersible in water without a particle aggregation due to the binding ability of the diphosphonate group to ferric ions. For the radiolabelling,^99m^TcO_4_^-^ was prereduced by SnCl_2_ and then mixed with Fe_3_O_4_-DP-PEG NPs in aqueous media under stirring for 30 min at room temperature. An ultrafiltration was performed three times to purify the NPs from the unbound radionuclides, obtaining a LE of 85.9%. The radiolabelling stability was evaluated by incubating the NPs in a foetal bovine serum (FBS) solution. Results showed a radiochemical purity of 93.8% after 6 h and 89.1% after 72 h [38]. NaGdF_4_:Yb,Tm NPs were synthesised by the thermal decomposition method and then labelled with ^99m^Tc through the LAGMERAL method, where the surface hydrophobic oleate ligand was replaced with DP-PEG-mal. The final LE was 42.6%. For Cu2−xS NPs, the same process of synthesis and radiolabelling were used, showing a final RCY of 89.5%. This work demonstrated that this new approach for the inorganic NPs radiolabelling does not alter their intrinsic physicochemical properties [39].

#### 2.1.2. Indirect Radiolabelling

In recent years, the indirect radiolabelling of NPs with ^99m^Tc has been reported by several studies aiming to create a more stable radiolabelling. Several bifunctional chelators (BFCs) have been studied, such as hydrazinonicotinamide (HYINIC), diethyl-ethylene-triamene-penta-acetic acid (DTPA), or the N-hydroxysuccinimide derivative of S-acetylmercaptoacetyltriglycine (NHS-MAG3) [6].

The NPs radiolabelling with HYNIC requires the presence of coligands, such as ethylenediamine-di-acetic acid (EDDA)/tricine, to stabilize the radioisotope. Due to the possible formation of ^99m^Tc-EDDA/tricine complexes, butanone is often used as the mobile phase in ITLC for quality controls. This solution allows the migration of free ^99m^TcO_4_^-^ to the front of the strips (RF: 1.0 for ^99m^TcO_4_^−^), while NaCl 0.9% as the mobile phase allows the migration to the front of both free ^99m^TcO_4_^-^ and ^99m^Tc-EDDA/tricine (RF: 1.0 for ^99m^TcO_4_^-^ and ^99m^Tc-EDDA/tricine). In this way, it is possible to discriminate the percentage of the radioactive EDDA/tricine complex in solution [40,41].

The HYNIC can be labelled with the isotope and then conjugated with NPs or can be firstly conjugated with the NPs and then radiolabelled. The last option allows to create stocks of conjugated NPs ready to use for the radiolabelling.

Both of these techniques were used to radiolabel gold NPs (AuNPs) with ^99m^Tc. Mendoza-Sánchez et al. used the complex HYNIC-Gly-Gly-Cys-NH2 (HYNIC-GGC) to perform the radiolabelling, where the HYNIC is the chelator of the isotope, -Gly-Gly- the spacer, and -Cys-NH2 is the active group that labels the AuNPs surface.

The conjugation was prepared by mixing the HYNIC-Gly-Gly-Cys-NH2 complex with the NPs solution at a molar ratio of 1:5000 (AuNPs:complex).

AuNPs, once purified from the free complex in solution with a size exclusion chromatography (SEC), were stored for 9 months at 4 °C. The authors demonstrated the high stability of HYNIC-Gly-Gly-Cys-NH2-AuNPs during the storage without the aggregation of NPs. These batches were radiolabelled with ^99m^Tc by using EDDA/tricine in 0.1 M phosphate buffer (pH 7) and 10 µg of and SnCl_2_ (HCl solution), followed by incubation at 100 °C for 20 min. Quality controls demonstrated an RCY higher than 95% and a high stability up to 24 h in human serum [42].

Estudiante-Mariquez et al. also labelled AuNPs that were previously conjugated with HYNIC.

After the conjugation, the unbound HYNIC was removed by ultrafiltration and the final number of HYNIC molecules per AuNP was estimated by a UV-Vis calibration curve of HYNIC. Results showed a high number of molecules associated to the NPs: 1199 ± 430 at 20 min, 2214 ± 146 at 2 h, 3063 ± 360 at 24 h, and 2989 ± 387 at 48 h.

With the ^99m^Tc previously reduced, it was added to the HYNIC-AuNPs solution and incubated for 20 min at 100°C under stirring. The purification of the mixture solution was performed by filtration using 100 kDa centrifugal filters at 12,000 rpm for 5 min. The LE and radiochemical purity were reported as 86% and 97%, respectively [43].

Morales-Avila et al. used the same complex, HYNIC-GGC-NH2, to radiolabel AuNPs, but with a different method. Indeed, they first performed the labelling of HYNIC-GGC-NH2 with the isotope by mixing an EDDA/tricine solution, SnCl_2_, and ^99m^Tc at 92 °C for 20 min in a dry block heater. This complex was then added to the AuNPs to perform the radiolabelling. The LE was verified with two different mobile phases in ITLC-SG: 2-butanone was used to determine the amount of free ^99m^TcO_4_^−^ (Rf = 1) and methanol/1 M ammonium acetate (1:1 *v*/*v*) for ^99m^Tc-colloid (Rf = 0). The results showed a LE higher than 95% [44].

Zhang et al. labelled the BFC, HYNIC, to hyaluronic acid-coated silver NPs (HA-Ag NPs), and then radiolabelled this complex using two coligands, tricine and the trisodium triphenylphosphine-3,3′,3″-trisulfonate (TPPTS). Results showed a high radiochemical purity (<98%) after purification by ultracentrifugation of the radiolabelled complex. However, the authors considered the presence of only two species in solution, the free ^99m^Tc and ^99m^Tc-HA-Ag NPs, excluding the presence of radiocolloids, which were not analysed in the ITLC. Moreover, after radiolabelling, the NPs exhibited slightly different hydrodynamic diameter and zeta potential values as compared to the unlabelled particles. SPECT imaging in mice revealed a strong signal in the abdominal area, particularly in the spleen and liver, indicating the uptake by RES [45].

The mercapto-acetyl-triglycine (MAG3) can be also used as BFC for ^99m^Tc-NPs radiolabelling. One of the advantages of this BFC is that no coligand is required, but it requires high temperature during the radiolabelling process, limiting this technique to molecules that are not temperature- or pH-sensitive [46]. Its application has been reported for the conjugation of antisense morpholino oligomers (MORFs) and successive labelling with isotopes. Usually, the oligomers are previously incubated with S-acetyl NHS-MAG3 and SnCl_2_, and heated at 100 °C for 20–25 min. Then, the MAG3-conjugated oligomers NPs are incubated with ^99m^Tc, with a final radiochemical purity over 95% [47,48].

DTPA is another BFC used as a chelating agent for ^99m^Tc, despite its chemistry being more favourable to isotopes such as ^111^In, ^90^Y, and ^177^Lu.

AuNPs were labelled with DTPA through the amines on the distal end of the polyethylene glycol (PEG) on the NP surface. A 10-fold excess of DTPA was used for the conjugation, then purified with one-day dialysis against saline solution. The complex was then radiolabelled with ^99m^Tc. SnCl_2_ were used as reducing agent and then added to the AuNPs suspension. The pH was adjusted to 7 and finally ^99m^Tc was added. The reaction vial was purged with nitrogen and stirred for 30 min at 80 °C under shaking. The final LE obtained was more than 90% [49].

Lee et al. conjugated DTPA to the amino groups of dopamine-functionalised SPIONs to perform the labelling with ^99m^Tc, thanks to the reaction between the reactive isothiocyanate group of DTPA and the amino groups of dopamine. The molar ratio of DTPA added to the NPs solution was 2:1 (DTPA:NPs). In the radiolabelling procedure, 46.3 MBq of ^99m^Tc in 0.24 mL of isotonic saline was added to 2 mg/mL of SPIONs solution, using 0.2 mg/mL of SnCl_2_ in 0.04 N HCl. The final mixture was incubated under stirring for 60 min at 37 °C. The LE, calculated by ITLC–silica gel strips in saline, was >95% [50].

Manganese-based mesoporous silica NPs were radiolabelled by Gao et al. with ^99m^Tc using DTPA as BFC. The radiolabelling process occurred firstly by conjugating DTPA to the NP surface and then by adding ^99m^Tc and SnCl_2_ in solution.

The RCY, calculated via ITLC, was approximately 99%. Moreover, the radiochemical stability of this radiopharmaceutical remained over 95% up to 24 h at 37 °C in FBS [51].

Helbok et al. compared two different radiolabelling methods for DTPA–liposome NPs: one using ^99m^Tc and SnCl_2_, and the other using the precursor [^99m^Tc(OH_2_)^3^(CO)^3^]^+^.

The LE and RCY were determined by ITLC using two mobile phases: 0.1 M citrate to calculate the percentage of the free isotopes (Rf 0.8–1.0) and a solution of pyridine/acetic acid/H_2_O (3/5/1.5) to calculate the percentage of radiocolloids (Rf 0.0–0.2).

The highest LE was obtained using the precursor, with a percentage of 98.45 ± 1.09%, compared to 74.85 ± 6.15% for direct radiolabelling [52].

The tricarbonyl precursor [^99m^Tc(OH_2_)^3^(CO)^3^]^+^ was also used to radiolabel silica NPs, with nitrilotriacetic acid (NTA) as chelating linker. This method involved three steps: prelabelling of the ^99m^Tc–tricarbonyl core, conjugation with poly-His peptide, and the labelling of the NPs. For the first step, a commercially available kit was used to prepare the labelling precursor ([^99m^Tc(H_2_O)_3_(CO)_3_]^+^). This kit contains 8.5 mg sodium tartrate, 2.85 mg sodium tetraborate decahydrate, 7.15 mg of sodium carbonate, and 4.5 mg sodium bicarbonate. The ^99m^TcO_4_^-^ was added to the kit vial and incubated for 20 min at 100 °C.

The second step consisted in the prelabelling of a hexa-His-Tag peptide, mixing the peptide solution with the (^99m^Tc[CO]_3_[OH_2_]_3_)^+^ the precursor solution at room temperature under stirring for 60 min. In the last step, the conjugation with NTA-NPs was performed, incubating the previously prepared solutions at room temperature under stirring for 2.5 h. The radiochemical purity calculated by high-performance liquid chromatography (HPLC) was greater than 95% [53].

DOTA is usually used for theranostic applications, labelling isotopes such as ^68^Ga, ^90^Y, ^177^Lu, ^225^Ac etc, but it is not widely used for the radiolabelling of NPs with ^99m^Tc.

However, Xing et al. radiolabelled PAMAM dendrimers through the acetylation of the terminal amines, allowing the DOTA chelation and the labelling with ^99m^Tc. The dendrimers were then used for entrapping AuNPs. The ^99m^Tc-pertechnetate solution was added to a vial containing SnCl_2_ and the NPs suspension was dissolved in PBS under continuous stirring. After incubation for 30 min at room temperature, the reaction was stopped, and the reaction mixture was purified by PD-10 desalting columns using PBS solution as mobile phase. The LE pre-purification was 64.5 ± 6.8%, raised above 99% after the purification [54]. Georgiadou et al. radiolabelled CoFe_2_O_4_ magnetic NPs with three different methods: (i) direct coupling through the octadecylamine (ODA) amino groups; (ii) indirect coupling through the amino groups of the intermediate linker polyethylenimine (PEI) molecules; (iii) loading on a PEG matrix after the PEGylation of the MNPs.

The radiolabelling was performed for all the batches incubating the ^99m^Tc and SnCl_2_ in the magnetic NPs suspension at room temperature for 1 h. The purification of the complexes was performed with a magnet. Quality controls performed with ITLC revealed an RCY of 31% for the first labelling approach, 34% for the second, and 48% for the third. All the batches resulted stable at room temperature for at least 24 h [55].

#### 2.1.3. Radiolabelling by Encapsulation

Encapsulating ^99m^Tc within the NPs can be performed with two different strategies: after or during the synthesis of the NPs.

The first approach requires the specific physical–chemical characteristics of the NPs to facilitate the encapsulation of the isotope into their core. For this reason, this strategy often requires the labelling of the radioisotope to a molecule which allows the encapsulation of this complex into the NP core.

Fluconazole was labelled with ^99m^Tc by simply mixing the drug with a solution containing the isotope and SnCl_2_. The LE of the complex was 94%. After an incubation under stirring at room temperature for 120 min, this solution was added to a solution of NPs. The encapsulation efficiency of the radiolabelled drug inside the NPs was approximately 30%, and 94% after purification with SEC [54]. A similar approach was used to radiolabel doxorubicin and trastuzumab, followed by their encapsulation in AuNPs and mesoporous silica NPs, respectively. AuNPs showed an RCY over 90%, with the following conditions: DOX concentration (1 mg/mL), SnCl_2_ concentration (100 μg/mL), reaction time (30 min), and pH (7) [56]. Mesoporous silica NPs showed a LE around 97% and a radiochemical purity over 95% up to 6 h [57].

Encapsulation of the radioisotope into the lipid NPs can occur using the lipophilic chelator, D,L-hexamethylpropyleneamine oxime (HMPAO). The ^99m^Tc-HMPAO complex can be easily obtained with a commercially available kit. This complex, once incubated with lipid NPs, can pass the NP layer and remain in their core after a reduction of HMPAO by glutathione. Solid lipid NPs (SLNPs) were radiolabelled by Videira et al. with ^99m^Tc by using the lipophilic chelator D,L-hexamethylpropylene amine oxime (HMPAO). HMPAO was reconstituted with ^99m^TcO_4_^−^ followed by the addition of SLNPs and an incubation of 10 min at room temperature. The labelling stability was assessed in vitro by monitoring the formation of suspensions in serum and plasma at 37 °C. The results showed a high LE (97%) and a loss of efficiency of approximately 5% at 4 h [58]. In the last years, the microfluidic technique for the synthesis of NPs showed interesting results in terms of batch-to batch reproducibility and encapsulation efficiency of several drugs [59]. This method has been used by Varani et al. to encapsulate ^99m^Tc in PLGA NPs during their synthesis. For the synthesis of PLGA NPs through the microfluidic system, two different phases are needed: an organic phase, usually acetonitrile, in which the polymers are dissolved, and an aqueous phase, where a stabilizer is dissolved. The mixing of the two phases in the microfluidic system under a constant flow allows the synthesis of NPs. By using this method, the encapsulation of the isotope and the synthesis of NPs occurred simultaneously. Two different approaches were tested: the addition of ^99m^Tc to organic phase and the addition of ^99m^Tc to the aqueous phase. Results showed no significant differences in terms of the LE (100% in both cases) after a purification with SEC. However, to obtain small NPs in terms of size, it was necessary to use a ratio that favoured the aqueous phase over the organic phase. Indeed, by increasing the volume of the organic phase with respect to the aqueous phase, the size of the NPs increased considerably. For this reason, when the isotope is added in organic phase, it needs to use a high amount of isotope considering the small amount of organic phase used for the synthesis of the NPs. Therefore, the RCY was lower compared to the method in which the isotope was added in the aqueous phase [60].

#### 2.1.4. Discussion

As discussed above, the direct labelling of ^99m^Tc with the NPs is a quick and easy process, but often the presence of a reducing agent is required.

The main limitation in the use of the reducing agent, such as SnCl_2_, is the formation of colloids after the radiolabelling process, which are very difficult to discriminate from the radiolabelled NPs. The production of colloids could be prevented with the use of other reducing agents, such as Na_2_S_2_O_4_, as described above_._

For the indirect radiolabelling of NPs, an ideal BFC should be able to create a stable complex of ^99m^Tc with a high yield at a very low concentration of the BFC-NPs conjugate. The BFC should also stabilize an intermediate or lower oxidation state of ^99m^Tc so that the complex is not subject to redox reactions and transchelation [61].

The surface labelling, with or without the BFC, could be appropriate for an easy formation of standardised kits used in clinical practice. The kit could be stored in NM departments and ready to use with a single-step radiolabelling. However, the isotope exposed on the surface of the NPs could be degraded by serum proteins in vivo. Nevertheless, the encapsulation of the isotope in the NP core can overcome this problem. In general, the radiolabelling of NPs with the ^99m^Tc offers the advantage of the easy availability of this isotope from generators routinely used in clinical practice. In addition, thanks to its chemical characteristics, it can be conjugated to different ligands or aminoacidic residues.

### 2.2. Radiolabelling with Indium-111

The long half-life of ^111^In (2.8 days) offers the opportunity to perform biodistribution studies or evaluate the therapeutic efficacy for a long time after the NP administration in vivo. For long-acquisition times of radiolabelled NPs, it is important to achieve a high LE, high in vitro and in vivo stability, and low interactions with biological molecules.

#### 2.2.1. Direct Radiolabelling

Direct radiolabelling of NPs with ^111^In have not been largely explored due to the need of high stability to perform prolonged studies in vivo. This stability often requires the use of chelators. Nevertheless, direct labelling does not expose the NPs to the excessive manipulations that are necessary with chelators. For this reason, a group of researchers established an innovative chelator-free method to radiolabel NPs. They used an ultra-small paramagnetic iron oxide nanoparticle (USPIONs).

USPIONs were labelled with ^111^In under aqueous conditions at pH 8.0, and the reaction mixture was heated and stirred at 120 °C for less than 1 h. Once returned to room temperature, the reaction was quenched adding an excess of DTPA, EDTA, or DFO.

This method showed great results in terms of the RCY (91 ± 2%) and radiochemical purity (>99%), with a low nonspecific binding (9 ± 4%) [62].

#### 2.2.2. Indirect Radiolabelling

Since ^111^In is a radiometal (class IIIB), the BFC during the radiochemical process is essential to have a stable radiochemical complex.

There are several BFCs used for the radiolabelling of NPs, such as 1,4,7-triazacyclononane-N,N′,N′′-triacetic acid (NOTA), 1-(1,3-carboxypropyl)-4,7-carboxymethyl-1,4,7-triazacyclononane (NODAGA), 1,4,7,10-tetraazacyclododecane-1,4,7,10-tetraacetic acid (DOTA), and 1,4,7,10-tetraazacyclododececane,1-(glutaric acid)−4,7,10-triacetic acid (DOTAGA).

All of these chelators require high temperature (40–60 °C) that may affect the stability of the NPs [3].

Banerjee et al. performed a two-step radiolabelling with ^111^In-DOTA to avoid the changing of physical properties of polymeric NPs after radiolabelling. The first step involved the radiolabelling of DOTA-PEG-alkyne with ^111^InCl_3_ at high temperature. In the second step, they performed the conjugation between ^111^In-DOTA-PEG-alkyne and azide-functionalised polymeric NPs at 4 °C. The long half-life of the isotope and the stability of the radiolabelled NPs allowed the visualisation of the tumour site up to 96 h post-injection [63].

Cheng et al. labelled mesoporous silica NPs with DOTA-N-hydroxysuccinimide-ester through amide formation at pH 4.5. The DOTA-conjugated NPs were then incubated with ^111^In in 2-(N-morpholino)-ethanesulfonic acid (MES) buffer at 60 °C, pH 5.5. The higher LE (95%) was obtained with a molecular ratio (DOTA/MSN) of 2000, maintaining a stability of 90% during the first 48 h [64].

Lumen et al. conjugated porous silicon NPs with DOTA-NHS ester, and then they radiolabelled this complex with [^111^In] InCl_3_ (2–10 MBq) in 0.2 M NH_4_OAc buffer (pH = 6.8, 37 °C) at 30 min with constant shaking. Results showed a radiochemical purity higher than 98% [65].

Hu et al. used a lipophilic chelate, 17 methoxy-DOTA-caproyl-phosphatidylethanolamine (MeO-DOTA-PE) to conjugate perfluorocarbon NPs. They used 0.5 M sodium citrate as a weak chelator in combination with ^111^InCl_3_ to minimize hydrolysis and precipitation of metal. The subsequent addition of Meo-DOTA-PE-conjugated NPs, a strong chelator, competed with the citrate to label ^111^In, resulting in a high yield of radiolabelling [66].

A different radiolabelling method was proposed by DeNardo et al. They conjugated the Chimeric-L6 (ChL6), human–mouse mAb chimera, to DOTA. The complex was purified and transferred in 0.1 mol/L ammonium acetate to perform the radiolabelling with ^111^In. Then, the compound was conjugated with SPIONs via amide linkage to the carboxyl (COOH)-terminated PEG coating [67]. SPIONs were also radiolabelled by Wang et al. but with a different BFC, the DTPA. Briefly, they incubated the BFC with the NPs and ^111^In in a citrate solution at room temperature, allowing the transchelation of ^111^In from the weak citrate complex to the DTPA complex [68].

DTPA was also used for the radiolabelling with ^111^In of several magnetic NPs.

DTPA-AuNPs were labelled with ^111^InCl_3_ in acidic buffer (pH 5–5.5), such as sodium citrate or sodium acetate, under heating (45 °C) with an incubation between 30 min and 1 h. Quality controls performed with ITLC showed a high LE over 95% [69,70,71]. The labelling of PLGA-NPs with DTPA is usually performed though a linker on the NP surface that allows the conjugation with the BFC.

Gill et al. firstly activated PLGA-NPs with 1-ethyl-3-(3-dimethylaminopropyl)carbodiimide/N-hydroxysuccinimide (EDC/NHS), followed by the conjugation with DTPA to human epidermal growth factor (hEGF). Then, this complex was radiolabelled with ^111^InCl_3_ in 0.1 M sodium citrate buffer (pH = 5.5) with a radiochemical purity >95% [72].

Wu et al. used the phospholipid phosphatidylethanolamine (PE) to perform the conjugation with DTPA. NPs modified with PE-DTPA were then incubated with ^111^InCl_3_ at 25 °C, with a final LE of 94.1% despite the low labelling temperature [73]. A low labelling temperature was also used to radiolabel AuNPs functionalised with DTPA. The resulting NPs were simply radiolabelled by adding ^111^In chloride and stirring overnight at 37 °C, obtaining a final high LE [74].

#### 2.2.3. Radiolabelling by Encapsulation

Despite the pros of encapsulation of the isotope inside the core of the NPs, a very small number of researchers have used this method to radiolabel NPs with ^111^In.

The encapsulation of radiometals, such as ^111^In, requires a carrier molecule that allows it to cross the NP surface. This approach it is usually applied for lipid NPs due to the difficulties of the metal radioisotopes to cross the lipid bilayer. Furthermore, a chelator is often entrapped inside the NP core that binds stably the isotope once it crosses the NP bilayer, avoiding the release of the radioisotope from the NP core.

The first example was reported by Gamble et al., with the encapsulation of ^111^In in lipid NPs using the antibiotic A23187, which allowed the transport of the radioisotope into the liposomal core. Inside the core of the liposomes, they entrapped the nitrilotriacetic acid (NTA) that chelated the radioisotope once inside the core, preventing its release and obtaining a final RCY of more than 90% [75].

Magnetic NPs usually encapsulate the ^111^In during their synthesis, resulting in an easy and fast method of radiolabelling.

This approach was firstly studied by Zeng et al., encapsulating the ^111^In in Fe_3_O_4_ NPs. Briefly, the radiopharmaceutical was prepared using pyrolysing ferric acetylacetonate (Fe(acac)_3_) in the presence of ^111^InCl_3_, α,ω-dicarboxyl-terminated polyethylene glycol (PEG), and oleylamine in diphenyl ether. Results showed a LE of 75% and high colloidal stability of NPs for 5 days in PBS and a water solution with a pH above 2. For solutions with a pH lower than 2, there was a massive release of the isotope due to the degradation of the NPs [76].

To protect the magnetic NPs from external conditions, avoiding their degradation or oxidation, Llop et al. encapsulated IONPs in a polymer shell. They first synthetised and radiolabelled IONPs in a single step, adding ^111^InCl_3_ during the coprecipitation of the magnetic nanomaterials. Second, they mixed the radiolabelled NPs with polymer nanomaterials during the emulsification process. The resulting NPs showed a high stability up to 48 h without isotope release. However, the authors did not show the LE and RCY of the radiolabelled IONPs, which should be important factors to consider when formulating a new radiopharmaceutical [77].

The intrinsic incorporation of the radioisotope was also applied for quantum dots (QDs), entrapping ^111^In in the crystal structure during their synthesis. Results showed the high values of the RCY (over 90%) [78].

Quinn et al. encapsulated ^111^In in AuNPs, avoiding the harsh radiolabelling procedures used for BFC-mediated radiolabelling. In this way, the AuNPs surface resulted free to conjugate the targeting sequence of arginine-glycine-aspartate (RGD). The efficiency of the labelling was higher than 95% [79].

#### 2.2.4. Discussion

Although there are few data in the literature, direct labelling with ^111^In avoids modifications in the NPs that can interfere with their physicochemical properties, thus modifying their biodistribution profile [80]. Coordination chemistry plays a significant role in BFC design, solution stability, radiolabelling kinetics, and the modification of pharmacokinetics. Due to its high charge density, ^111^In prefers hard donors such as N-amine and N-carboxylate atoms. For this reason, the BFC DOTA is the most used. Indirect radiolabelling may require high temperatures that could cause the degradation of the NPs during the radiolabelling process. Therefore, the selection of NPs and methods is crucial when planning the radiolabelling [81].

The encapsulation of radioisotopes during NPs synthesis represents an efficient method of radiolabelling, especially for magnetic NPs. However, this method could have some limitations for clinical applications, including radiation exposure to the operator during NP purification after synthesis. The pros and cons of the different methods for radiolabelling NPs with ^111^In are shown in Table 1.

### 2.3. Radiolabelling with Iodine-125 and Iodine-131

Iodine, and in particular ^125^I and ^131^I, has also been investigated for labelling NPs for SPECT imaging.

The ^125^I has been mostly used for in vitro studies due to the long half-life (60 days) and low-energy gamma emission.

The ^131^I is, instead, used for the radiolabelling of NPs for theranostic purposes. Indeed, ^131^I, with a physical half-life of 8 days, decays with gamma ray emissions, used for SPECT/CT imaging, and the emission of beta particles, used for therapeutic applications [82].

The radiolabelling of NPs with iodine radioisotopes occurred with halogen nucleophilic exchange based on an oxidising agent by direct radioiodination of NPs or indirectly, using prosthetic groups on NPs surface [83].

To allow the nucleophilic exchange, tyrosine residue is first coupled to the NPs, then the orthoposition of its phenolic group is electrophilically substituted by radioiodine. This process is fast, taking only a few minutes [84].

The N-chlorobenzenesulfonamide (Chloramine-T) is used in solution as a strong oxidant that triggers the radioiodination in a few seconds [85]. Then, a reducing agent (usually sodium metabisulphite) must be used to stop the reaction. This method is fast, low-cost, and reproducible, but the oxidant or reducing agents can affect the biomolecules present in solution. To control the oxidising activity of Chloramine-T, it is often fixed in polystyrene beads, thus the oxidising reaction can be stopped simply by removing the beads from the solution. A less aggressive labelling process is provided by using 1,3,4,6-tetrachloro-3α,6α diphenylglycouril (Iodogen). Iodogen is dissolved in organic solvent and evaporated in the presence of a tube, remaining bound to the wall. This prevents its dissolution in solutions and its interactions with the bioactive molecules/NPs.

#### 2.3.1. Direct Radiolabelling

Iodine radioisotopes can be directly labelled to the NP surface via chemical bonds between the nanomaterials and the radionuclide.

Halogen radioisotopes, such as ^125^I and ^131^I, easily bind metal nanomaterials due to the strong chemical affinity between metal and nonmetal materials.

The ^125^I-radiolabelled AuNPs are generally performed by simply mixing the isotope with the NP suspension. This is due to the high affinity of gold atoms with these ions, allowing an easy and stable labelling by chemisorption with final the LE and RCY over 90% [86,87,88].

Magnetic NPs can also be radiolabelled with the use of Iodogen as oxidising agent. AuNPs were radiolabelled by Su et al. by adding Na^125^I (370 MBq) and Iodogen (20 μg) into the NP solution. The results showed a LE of 86.3 ±  2.4% and a radiochemical purity greater than 99% after centrifugation [89].

The presence of Chloramine-T as an oxidising agent for ^125^I-radiolabelling has been used for other magnetic NPs [90]. AgNPs were radiolabelled by Chrastina et al. using Chloramine T as the oxidising agent. The oxidising agent and the carrier-free Na^125^I was incubated at room temperature for 10 min. The solution was then added to the AgNPs suspension in 10 mM HEPES, pH 6.0, under rapid agitation with a vortex. Free isotope in solution was purified with a size exclusion chromatography (G-25), and the final RCY was above 80% with a SA of 0.4–0.6 μCi/μg of ^125^I-Ag NPs [91].

Farrag et al. studied a different oxidising agent for the radiolabelling of AgNPs. They incubated the isotope (carrier-free Na^125^I) with different concentrations of Chloramine-T, Iodogen, and 1-Bromo-2,5-pyrrolidinedione (N-Bromosuccinimide) at room temperature for 10 min. The highest LE was obtained by using Chloramine-T and Iodogen as oxidising agents. Indeed, N-Bromosuccinimide is usually used for the radioiodination of peptides and proteins due to its mild effect as an oxidising agent [92]. For the radiolabelling of AgNPs, Fayez et al. investigated the influence of the reaction parameters to improve the radioiodination efficiency. They investigated the Chloramine-T amount, the pH, the reaction time, and the volume of the NPs solution. The highest RCY (97%) was obtained by mixing 15 μL (30 μg) of Chloramine-T with 20 μL of Na^131^I (4.1MBq) solution and 0.5 mL of AgNPs for 30 min at room temperature with a pH of 5 [93].

Gold nanorods (GNR) were radiolabelled by Zhang et al. with a simple one-step procedure, incubating [^131^I]NaI directly with the NPs. This labelling is possible thanks to the strong affinity of iodide ions to gold nanomaterials. The authors demonstrated how this method resulted in a high LE and radiochemical purity (over 95%) [94].

A similar procedure was performed by Liu et al. by using another metallic nanomaterial with the same characteristics of AuNPs. Copper sulphide NPs also have a high affinity for iodide ions, allowing for the labelling with a simple incubation with ^131^I in sodium perchlorate at 32 °C for 30 min [95].

Sakr et al. reported the same procedure for the radiolabelling of AgNPs. Briefly, they incubated silver nitrate with PEG and ^131^I in aqueous solution under stirring for 5 min, leading to a LE of 98% [96].

#### 2.3.2. Indirect Radiolabelling

During a typical process of radioiodination with the indirect method, the biomolecule-coated NPs are incubated with radioactive iodide in a slightly alkaline buffer (pH 7.5) and an oxidising agent. The oxidant agent oxidizes I^-^ in I^0^, which is able to bind several prosthetic groups as tyrosine residues on the NP surface. After the radioiodination, an excess of a reducing agent, such as sodium metabisulfite, is added to the solution to inactivate the oxidation process [97]. Tang et al. radiolabelled IONPs with ^125^I by the radioiodination of amino groups on the NP surface with the Iodogen oxidation method. Indeed, they first modified the NPs with 3-aminopropyltriethoxysilane and with N-succinimidyl-3-(trinbutyl stannyl) benzoate to introduce amino groups on the surface of the IONPs. Then, carrier-free Na ^125^I (1 mCi) was added together with the NPs suspension in Iodogen-coated glass vials and vortexed for 5 min. This method showed a high LE (93%) and high stability (96% of ^125^I retained up to 15 days) [98]. The dimeric cRGD peptides [cyclic(Cys-Arg-Gly-Asp-dSer-Cys)-Tyr-dSer-Lys-Tyr-cyclic(Cys-Arg-Gly-Asp-dSer-Cys)] contain a tyrosine amino acid which can be easily labelled with ^125^I. First, the NPs were functionalised with the (cRGD)_2_ peptide, then, through the standard Chloramine-T method, were labelled with the radioisotope, followed by few minutes of incubation under shaking, obtaining a LE over 99% and a stability above 90% at 24 h [99,100]. Another peptide used for the conjugation of AuNPs was pMMP9 (sequence: DTPA-Gly-Pro-Leu-Gly-Val-Arg-Gly-Lys-Gly-Tyr-Gly-Ahx-Cys-NH_2_), which contains a cysteine residue to anchor to the gold surface, a tyrosine residue to label the ^125^I, and DTPA to label the radiometals. Due to these residues, it was possible to radiolabel the AuNPs with both ^111^In and ^125^I in two successive steps. First, ^111^InCl_3_ was added to the NPs in an acidic buffer at 45 °C and incubated for one hour, then the reaction mixture was resuspended in PBS buffer and centrifuged to remove the unbound ^111^In. The pellet was then added to an Iodogen tube and incubated with Na^125^I for one hour. The radiochemical purity was evaluated with ITLC and resulted in more than 95% [70].

Monoclonal antibodies (mAb) were also used to allow the labelling between the isotope and NPs. The group of Liu chose the antigastric cancer mAb 3H11 conjugated on the Fe_3_O_4_ nanocrystals surface. They used two strategies: in the first approach, mAb 3H11 was first conjugated to the NPs, then labelled with ^125^I through Iodogen method. In the second strategy, the mAb was first labelled with the radioisotope and then conjugated to the NPs. The resulting effectiveness was nearly the same, but in the first strategy was required a purification step before the radiolabelling process to remove impurities introduced by the conjugation reaction [101].

For the radiolabelling with ^131^I, Zhao et al. used single-walled carbon nanotubes (SWNTs) coated with a shell of polydopamine (PDA) and subsequently modified by PEG. The radiolabelling process was performed via the Chloramine-T method, allowing the bond of the benzene rings in polymerised dopamine through the electrophilic substitution reaction, achieving a high RCY (~90%) [102,103]. AuNPs and dendrimers were labelled with ^131^I through the bound between the radioisotope and the phenol group of HMPAO previously conjugated. Both the radiolabelled NPs showed a good stability and high radiochemical purity [104,105].

#### 2.3.3. Discussion

The methods described for iodine labelling mainly regard ^125^I that is suitable for in vitro studies but not for imaging. Therefore, we cannot assume that direct labelling methods will be efficient for ^123^I as well as they are for ^125^I.

On the other hand, the use of ^131^I has been applied only to metallic NPs with little experience as compared to ^111^In or ^99m^Tc-labelled NPs, and mainly to label biomolecules on the surface of NPs. Poor data are available on the possible modifications on the structure of these radiolabelled biomolecules, being a limitation in the use of radioiodine, as shown in Table 1.

## 3. General Conclusions

This review highlights the various methods for radiolabelling, with different radioisotopes, NPs for their diagnostic use in SPECT. Today, nanomedicine represents one of the most promising technologies in the biomedical research field. In NM, its application has seen a huge increase in research as drug a delivery system or as imaging probes.

Multiple approaches may be possible by combining NPs and radioactive isotopes, allowing for multimodal therapy, such as the encapsulation of a drug plus a therapeutic isotope, multimodal imaging, such as magnetic NPs or other contrast agents plus a diagnostic isotope, or a theranostic approach, which combines a diagnostic and a therapeutic isotope. Each specific application of NPs is based on the most appropriate choice of nanomaterials, isotope, and the most suitable radiolabelling method. Indeed, all these factors influence the result and the possible translation in humans. The presence of a wide choice of nanomaterials and isotopes has led to several methods for their radiolabelling.

The physicochemical properties of NPs can be such as to allow a direct labelling between the isotope and the NPs without spacers or chelating agents. For example, the high affinity between the radioiodine and metallic NPs, such as AuNPs, CuNPs, or AgNPs, allows a chemical absorption with a high LE and excellent stability in vitro and in vivo.

Direct labelling does not require a further manipulation of NPs, avoiding a possible change in the physicochemical characteristics of NPs, resulting in a different biodistribution in vivo.

However, several isotopes (e.g., radiometals) require the presence of a BFC, such as NOTA, DOTA, NODAGA, HYNIC etc., to perform an efficient radiolabelling with a high stability.

The radiolabelling through a BFC often requires working at high temperatures to facilitate the conjugation of the BFC to the NPs. This could represent a limit to their use, as it could alter the physical characteristics of the NPs.

Although NPs showed promising results in NM, some important issues regarding the radiochemical procedures should be standardised for successful translation into clinical applications. These aspects would warrantee a larger applicability of NPs, not only for diagnostic purposes able to achieve a personalised diagnosis, but also for planning tailored therapies. The larger application of nanomedicine would, indeed, offer in the future a novel possibility to approach many clinical scenarios, in particular in the oncological field.

## Figures and Tables

**Figure 1 biomolecules-12-01522-f001:**
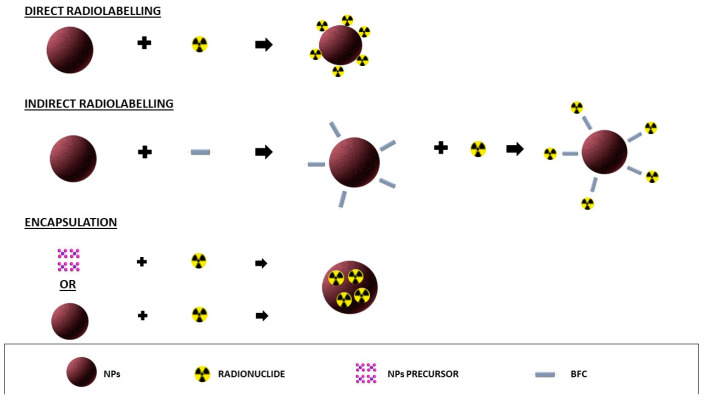
Different radiolabelling methods of NPs.

**Table 1 biomolecules-12-01522-t001:** Pros and cons of different methods for radiolabelling NPs.

Method/Isotope	Advantages	Disadvantages	Indications	Improvement
Direct labelling with ^99m^Tc	Cheap and easy labelling process	The use of SnCl_2_ as reducing agent may induce the formation of colloidal particles	Evaluate pH and incubation time for an optimal radiolabelling	Different reducing agents can avoid the colloidal formation
Indirect labelling with ^99m^Tc	Stable radiolabelling, incubation at room temperature	May dissociate in vivo due to interaction with proteins	The use of a reducing agent may be necessary	Stabilise an intermediate or lower oxidation state of ^99m^Tc so it is not subject to redox reactions
Encapsulation of ^99m^Tc	Possible during the NP synthesis. No direct contact between isotope and proteins in vivo	Requires isotopes with long half-lives compared to the time of synthesis	Consider the method of synthesis or NPs properties to have high efficiency	Select the best method of synthesis to have reproducible results
Direct labelling with ^111^In	Avoid modifications on the NPs surface with BFC	Requires high temperature	Thermodynamic and kinetic stability studies are significant	Overcomes the limitation of the selection of chelators
Indirect labelling with ^111^In	Allows high stability in vitro and in vivo for long acquisition times	Requires high temperature	Usually requires surface modification	Better use BFCs with polydentate chelators with hard donors, such as amine-N and carboxylate-N atoms
Encapsulation of ^111^In	Prevents the dissociation of radionuclide from NPs in vivo	Excessive exposition to radiations for the operator	Consider the method of synthesis or NPs properties to have high efficiency	Defined the best protocol of synthesis/radiolabelling for each type of NP
Direct labelling with ^125^I/^131^I	Easy to perform and cheap	May induce oxidation of NPs and biomolecules	^125^I-labelled NPs only for in vitro studies. ^131^I-labelled NPs for theragnostic applications	Better to rely on indirect labelling method for reproducibility and biochemical characterisation
Indirect labelling with ^1125^I/^131^I	Reproducible	Time-consuming and requires more expertise	^125^I-labelled NPs only for invitro studies. ^131^I-labelled NPs for theragnostic applications	^123^I can be used for SPECT imaging and ^124^I for PET imaging

## Data Availability

Not applicable.

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
