# Peer review of "Methods for Radiolabelling Nanoparticles: SPECT Use (Part 1)"

_biomolecules, 2022, doi:10.3390/biom12101522_

Round 1

Reviewer 1 Report

The authors reviewed in detail the methods of radiolabeling nanoparticles with several radionuclides commonly used in SPECT in the manuscript. The radiolabeling methods have been introduced according to the nuclide types, i.e., 99mTc, 111In and 125/131I, followed by a discussion relevant in each section. Therefore, the readers could quickly find the methods for a certain nuclide. As the finishing touch, the table gives the readers very clear information about the pros and cons of the various methods. However, the logic of the whole manuscript is slightly confusing, and it is recommended to be published after addressing following major concerns.

1. When introducing the methods, especially, the logic is confusing for part 2.1.1. It is suggested to classify and describe the methods according to the different labeling principles, and discuss the differences, advantages and disadvantages of various methods. The current narration is like enumerating without distinction, and there is no rules and regulations for segmentation.

2. The current manuscript is more like a simple summary of experimental methods, lacking a necessary description of the labeling principle. Additionally, it is suggested to add the discussion about the impact of labeling methods on the properties of nanoparticles.

3. According to the author's classification, the method described in lines 293-316 should be classified as direct labeling as it does not involve the bifunctional chelators, and the labeling principle was studied and discussed in detail in Small, 2021, 17(51), 2104977.

4. Why is the section 2.3 not classified as usual? The radiolabeling methods for 125/131I can also be divided into direct radiolabeling and indirect radiolabeling with prosthetic groups.

5. It is suggested to add some necessary figures to facilitate quickly understanding of relevant chemical structures and labeling principles.

6. There is a lack of detailed description or explanation in many places. For examples, what is the optimal pH and time in lines 120-123? How many HYNIC molecules are there on each AuNPs in lines 230-232? How about the labeling rate and stability of this method in lines 252-255? Why the RCY is higher for the radiolabeling approach with the addition of the isotope in the aqueous phase in lines 355-356? Without these, the corresponding content of the original manuscript seems to be meaningless.

7. For the encapsulation of 111In, Zeng et al. and Llop et al. also contributed very valuable methods for Fe3O4 nanoparticles (Chem. Commun., 2014, 50(17), 2170-2172; J. Mater. Chem. B, 2015, 3, 6293), as well as Sun et al. for quantum dots (J. Mater. Chem. B, 2014, 2, 4456), which are recommended to be carefully reviewed.

8. There are some errors in the manuscript. For example, lines 56-57, the labeling method using chelators should not be considered as direct labeling. Line 309, the word LE should be replaced by radiochemical purity. There are also many further and grammar errors, and it is recommended to carefully check, modify and polish the manuscript. Additionally, the introduction section is suggested to be segmented for easy reading.

9. The references are relatively balanced. However, the authors still can refer to and include latest reviews in the field (ACS Appl. Nano Mater. 2022, 5, 7, 8680-8709; Chem. Soc. Rev., 2021, 50, 3355-3423; VIEW, 2020, 1(2), e19; Biomaterials, 2020, 228, 119553).

Author Response

File attached.

Reviewer 2 Report

The manuscript describe, analyze and compare several methods for radiolabeling nanoparticles.

An adequate review of radiolabeling methodologies and techniques is shown.

I believe that it is necessary to limit the definition of nanoparticle, since the limits of the review depend on it. Occasionally, some types of lipid and polymer micelles are considered nanoparticles, the review is limited to a few examples of inorganic and polymer particles, without addressing the current status of radiolabeling of polymer and lipid nanoparticles with theranostic applications (including diagnostic by spect).

It is necessary to improve the review of the state of the art, at least in the scientific advances in the last three years. Previous reviews have been reported that cover part of the subject.

In addition to radiochemical purity, radiolabeling efficiency is a parameter that determines the quality of the formulation. What is the contribution and observations of the authors regarding this parameter?

and on the specific activity of radiolabelling of nanoparticles?

Author Response

File attached

Round 2

Reviewer 1 Report

There are still many problems raised before in the current version, and it is recommended to be published after addressing them:

1. The logic of section 2.1.2 in the revised version is still confused, and the author's narrative is a bit of a departure, as the focus should be the labeling method of nuclides, rather than how to modify chelators on nanoparticles.

2. In the revised manuscript, the segmentation remains absent-minded, and the writing format and grammar do not seem to be taken seriously.

3. Some recommended references do not appear in the revision.

4. It is suggested that in the reply, the location of the revision should be indicated for convenience.

Author Response

Replies enclosed

Round 3

Reviewer 1 Report

The authors addressed all the concerns raised. I recommend its publication at the current form. 

Author Response

Thanks